# Prevalence and consumption patterns of energy drinks among Iraqi adolescents: A cross-sectional study

**Soran Abdullah Al-keji**[‡*], **Samir Mahmood Othman**[‡]

Department: Community Medicine, College of Medicine, Hawler Medical University, Erbil, Iraq

‡ SA performed the main analysis and drafted the manuscript. SM Supervised the study and critically revised the manuscript. Both authors approved the final version.
* soranalkeji@gmail.com

## Abstract

### Background

Most energy drinks contain significant levels of caffeine, sugar, and other ingredients, which are consumed increasingly among adolescents. Excessive energy drinks consumption is associated with various adverse health effects.

### Objectives

To investigate the prevalence and pattern of energy drink consumption among high school students in Erbil City, The Kurdistan Region, Iraq.

### Methods

A cross-sectional study was conducted from October 15th, 2024 to March 15th, 2025. A multistage cluster sampling technique was used to recruit 800 students from 20 selected high schools in Erbil, Iraq. SPSS version 27 was used for data entry and analysis, a p-value ≤0.05 is considered statistically significant.

### Results

Overall, 82.1% of participants had reported at least a one-time-consumption of energy drinks in their lives, and 57.6% had consumed them in the past 30 days. The prevalence of energy drink consumption was significantly higher among males (58.3%, p = 0.001). The study found a statistically significant association between gender (p = 0.001) and living arrangement (p = 0.023), and the prevalence of energy drink consumption.

### Conclusion

The prevalence of energy drink consumption among high school-aged adolescents in Iraq was high, and associated with sociodemographic characteristics. The finding

**Data availability statement:** All relevant data are within the paper and its Supporting information files.

**Funding:** The author(s) received no specific funding for this work.

**Competing interests:** The authors have declared that no competing interests exist.

highlights the importance of considering gender differences in understanding energy drink consumption patterns. The study underscores the need for educational interventions to enhance students' awareness of the health risks associated with energy drink consumption.

## Introduction

Energy drinks are available in more than 140 countries, contributing to a multibillion-dollar industry globally, that was initially introduced in Austria in 1987 [1]. Energy drinks are advertised to increase energy, reduce symptoms of fatigue, and improve mental alertness and concentration. They contain significant levels of caffeine and sugar along with additional compounds such as B vitamins, amino acids, and herbal stimulants like guarana [2].

Several adverse health effects are associated with excessive energy drink consumption [3]. The acute health effects include irritability, nervousness, tremors, insomnia, nausea, anxiety, cognitive disorders, diuresis, tachycardia, arrhythmia, fever, gastrointestinal disorders, elevated respiratory rate, and reduction of myocardial blood flow. Moreover, long-term exposure to caffeine is associated with severe health effects including a range of disfunctions of the gastrointestinal, muscular, and renal systems as well as the liver [3,4]. Energy drink consumption has significantly increased during the last few decades, especially in Western and Asian nations. Youth-oriented media is frequently used to promote energy drinks [5].

Recent evidence shows growing public concerns about energy drink consumption among teenagers and young adults. The components of energy drinks are often unknown to teenagers and young adults. These beverages contain high amounts of caffeine and other substances that are often not present in food; energy drink consumption, especially when consumed repeatedly, can result in possible adverse health consequences to the general public, especially to adolescents [6].

Students are a segment of the population that is very susceptible to stress, which is mostly caused by their lifestyles, studies, and tests. Thus, they are more likely to consume energy drinks containing caffeine and other unusual ingredients. Even more concerning, the majority of them are unaware of these drinks' harmful adverse effects. The energizing, deceptive effects of energy drinks make people feel more alert than they actually are. [7]. The most commonly acknowledged reasons for consuming energy drinks include studying for an examination, preparing a project, and staying alert. Previous studies have found a positive correlation between perceived stress and energy drink consumption; this association between academic performance, perceived stress, and energy drink consumption suggests that younger students may become more dependent on energy drinks [8].

Consumption of energy drinks has grown dramatically in the recent years. Teenagers, athletes, and students are among the most frequent customers for various reasons, including social influence, taste preference, performance

enhancement, and marketing strategies [9,10]. Despite their widespread use, only a few studies have examined the long- term effects of these beverages on health [10]. Energy drink consumption is highly prevalent among adolescents and school students, and has become a topic of interest in recent decades, with varying rates across countries worldwide. The lowest prevalence was reported from Finland, 24.4% [11], and 31.3% in Hungary [12], and the highest prevalence were reported in Congo, 63% [13], and 66% in America [14]. In the Gulf Countries, which have a similar cultural and sociodemographic relation to Iraq, there was a prevalence of 45–60% among both middle and secondary school students [15]. In Iraq, only one study has been conducted across different Middle Euphrates Cities; participants included college students, fitness center users, and public place attendants, but not high school students, where the prevalence of monthly energy drink consumption was 50.2% [16]. Bearing in mind, the large number of high school students in Iraq and the several types of affordable and readily available options for them underscores the need for an estimation of the prevalence of consumption among high school students. Especially, its relationship with the educational background of the parents and the sociodemographic factors may be affecting the consumption of energy drinks among adolescents in Iraq.

There is limited information available on energy drink consumption among high school students in Erbil, the capital of the Kurdistan Region of Iraq. This study aims to assess the prevalence and consumption patterns of energy drinks among high school students in Erbil City, Kurdistan Region of Iraq.

## Materials and methods

### Study design, sample size, and sampling method

A school-based cross-sectional study was conducted in Erbil, Iraq, from October 15th, 2024, to March 15th, 2025. Before starting data collection, the consent form was obtained from the schools' administrations, and convenient dates were scheduled. Epi Info version 7 was used to estimate the sample size by using a confidence interval of 95%, 5% degree of precision, estimated frequency of 42.7% from the other study done in the region [7], population size of 29013, and design effect 2; the estimated sample size was 732. A multistage cluster sampling technique was used for selecting schools and students. In the first stage, a sample of 800 high school students from 20 schools out of 72 high schools in Erbil city by using the probability proportional size sampling technique. In the second stage, a stratified random sampling method was used to select students in the selected schools. The researchers developed the questionnaire based on previously published studies on energy drinks. It involved closed-ended questions, with some items adapted from earlier questionnaires, and minor modifications made to align with the objectives and context of the current study to assess the prevalence and consumption patterns of energy drinks [17,18]. The questionnaire included (10 items) on socio-demographic characteristics and (4) on energy drink consumption patterns.

### Data collection tool

Before the data collection process, the draft questionnaire was reviewed by three experts in epidemiology and public health to assess its content relevance, clarity, and alignment with the study's theoretical construct. Based on experts' feedbacks, several items were revised, refined, or reworded to ensure that each question captured their intended constructs and minimized ambiguity. These expert reviews strengthen the questionnaire's content and construct validity and enhance the instrument's reliability before final administration. Additionally, a pilot study was conducted at a school not included in the sampling frame, involving 20 students, to evaluate feasibility, item clarity, and data collection procedures. Feedback from the pilot study informed minor modifications to the questionnaire, thereby improving reliability and validity before the main study. The questionnaire included closed-ended and multiple-choice questions, organized into two sections. The first section included the sociodemographic characteristics. The second section evaluated energy drink consumption patterns among participants. Questions related to energy drinks included consumption patterns and frequency

of consumption. The questionnaire was designed to address the study's questions related to the aim and objectives, initially in English and translated into the local languages (Kurdish and Arabic).

## Ethical considerations

Prior to starting the study, the ethical approval was obtained from the College of Medicine Ethics Committee at Hawler Medical University, Kurdistan Region, Iraq (Code 9, September 2024). Before data collection, verbal informed consent was obtained from all students who participated in the study, and participants were informed that participation in this study is entirely voluntary and that they had the right to withdraw at any time without penalty. The study involved minimal risk, as an anonymous and self-administered questionnaire was used, in accordance with the International Ethical Guidelines (2016) and the Declaration of Helsinki [19,20]. Ethics committees may approve alternative consent processes, including verbal informed consent without written parental consent. The consent procedure was previously reviewed and approved by the Ethic Committee. Verbal consent was obtained after a clear explanation of the study and its objectives, witnessed by teachers from selected schools, and documented in a consent log sheet. The study did not involve any medical experimentation, and participants were also notified that the questionnaire did not include any sensitive topics and stigma. In addition, their replies would be kept anonymous, and that the obtained data would be used solely for research purposes. The Directorate of Education in Erbil had recieved permission to conduct the study. Students completed the questionnaire voluntarily and independently after receiving standardized instructions about the study's purpose and procedures.

## Statistical analysis

Data were entered into a computer using the Statistical Package for the Social Sciences (SPSS, version 27). Descriptive statistics were used to calculate frequencies and percentages, including life time prevalence, and previous month prevalence. Figueres were presented as a pie chart to depict the distribution of the energy drink consumption visually. Analytical statistics employed Pearson's Chi-square test to analyze the association between categorical variables such as the socio-demographic characteristics and energy drink consumption prevalence. The socio-demographic characteristics that showed an association at $p < 0.05$ in the chi-square analysis test were subsequently entered into multivariate logistic regression method to identify variables independently associated with energy drink consumption. The adjusted odds ratios (AORs) with a 95% confidence intervals (CIs) were reported. A p-value $< 0.05$ indicates statistical significance.

## Variable and measurements

The primary outcome of the study was energy drink consumption, defined as self-reported intake. Participants were classified as energy drink consumers (coded as 1) or non-consumer (coded as 0). Age, gender, academic level, living arrangement, religion, and ethnicity were among socio-demographic characteristics assessed. living arrangements and religion were included as sociodemographic variables in the study; they may be associated with adolescent's health behavior through family context, cultural norms, and parental supervision level. The education level of the head of the family, family income, car ownership, house ownership, and occupation of the head of the family were used to determine the socio-economic status of the participants by using a validated composite measure designed for health research in Iraqi society, by Omer Al-Hadithi with a predefined cutoff point used to classify the socio-economic status into low, medium, and high [21]. Scores for each participant were summed and categorized as low (0–3), medium (4–6), and high 7–10). The study population was homogenous in age, consisted of students aged 15–20 years, and who were not financially responsible for their household; consequently, the age component was excluded from the original index. A structured self-administered questionnaire was used to collect data. Variables included in the multivariate regression, such as gender, education level of the head of the family, living arrangement, and socio-economic status were selected based the results of the chi-square bivariate analysis.

## Results

### Socio-demographic characteristics of the study sample

Table 1 shows the socio-demographic characteristics of the research participants. In terms of the academic level of the participants among the 800 students who completed the questionnaire, the highest proportion belonged to 10th grade (379 students, 44.2%), the lowest was 12th grade (191 students, 21.7%), in comparison the rest were 11th grade (251 students, 28.5%). The study sample had a slightly higher proportion of female students than male students. Females comprised of 57.3% (n = 458) of the participants, while males accounted for 42.7% (n = 342). The age of the high school students in the selected sample ranged from 15 to 20 years, with a mean age of approximately 16.5 years and a standard

**Table 1. Socio-demographic characteristics of the study participants (n = 800).**

| Socio-demographic characteristics | | No. | % |
|---|---|---|---|
| Academic Level | Grade10 | 379 | 47.4 |
| | Grade11 | 241 | 30.1 |
| | Grade12 | 180 | 22.5 |
| Sex | Male | 342 | 42.8 |
| | Female | 458 | 57.3 |
| Age in years | 15 | 184 | 23.0 |
| | 16 | 234 | 29.3 |
| | 17 | 222 | 27.8 |
| | 18 | 115 | 14.4 |
| | 19 | 34 | 4.3 |
| | 20 | 11 | 1.4 |
| Religion | Muslim | 795 | 99.4 |
| | Christian | 3 | .4 |
| | Kakayi | 2 | .3 |
| Living Arrangement | With both parents | 760 | 95.0 |
| | with a single parent | 30 | 3.8 |
| | Others | 10 | 1.3 |
| Ethnicity | Kurdish | 755 | 94.4 |
| | Arabic | 23 | 2.9 |
| | Turkish | 22 | 2.8 |
| Education level of the head of the family | Illiterate | 44 | 5.5 |
| | Able to read and write, Primary | 353 | 44.1 |
| | Intermediate school | 96 | 12.0 |
| | High school or vocational | 78 | 9.8 |
| | Bachelor's degree (college) and above | 229 | 28.6 |
| Family income | Insufficient | 140 | 17.5 |
| | Sufficient for daily needs | 431 | 53.9 |
| | Exceeds daily needs | 229 | 28.6 |
| Occupation of the head of the family | Unskilled workers | 101 | 12.6 |
| | Lower professional | 516 | 64.5 |
| | High professional | 183 | 22.9 |
| Socio-economic status | Low | 80 | 10.0 |
| | Medium | 439 | 54.9 |
| | High | 281 | 35.1 |

deviation of 1.191, indicating that most participants' ages were clustered reasonably close to the mean. The majority of the participants were 16 years old, accounting for 29.3% (n=234) of the total study sample, and the least age group was 20 years old, with only 1.4% (n=11) of the sample.

Regarding participants' religions, the vast majority of the respondents were Muslim, accounting for 795 students (99.4%). The sample's living arrangement showed that a substantial majority, 95% (n=760), reported living with both parents. A small proportion of participants lived with a single parent, 3.8% (n=30), while only 1.3% (n=10) lived with others.

The distribution of participants by ethnicity indicates that the predominant proportion of respondents, 94.6% (n=755), were Kurdish. A small proportion of participants were Arab, 2.9% (n=23), and Turkish, 2.8% (n=22), respectively. The education level of the head of the family in the study sample showed that the most common level was completed primary school 44.1% (n=353), and fewer participants reported that their head of the family had completed high school (9.8%) or was illiterate (5.5%).

Moreover, the families' monthly incomes showed that 53.9% (n=431) of the respondents stated that their family's monthly income was sufficient for daily needs, and only 17.5% (n=140) reported that it was not sufficient for their daily lives. These results suggested that most of the participants lived in a family with at least a basic level of economic stability. The occupation of the head of the family of the participants showed that 64.5% (n=516), of the study sample, stated that their household earners worked in lower professional roles, and 12.6% (n=101) came from households where the main occupation was unskilled work. This distribution of the data indicated that most of the study participants came from middle-level occupational backgrounds.

The distribution of the study sample by socio-economic status demonstrated that the majority of the participants, 54.9% (n=439) were classified as having a medium socio-economic status. After that, 35.1% (n=281) fell within the high-level category, while only 10% (n=80) were categorized as the low level. This distribution indicates that a smaller proportion of the participants lived in low-income conditions, and most of the respondents were from middle-to-upper income backgrounds.

## Lifetime prevalence of energy drinks consumption

Fig 1 demonstrates the lifetime prevalence of energy drink consumption among the study sample. The majority of participants, 82.1% (n=657), reported consuming energy drinks at some point in their lives, in comparison, only 17.9% (n=143) stated that they had never consumed energy drinks. This result suggested a relatively high lifetime prevalence of energy drink consumption among high school students.

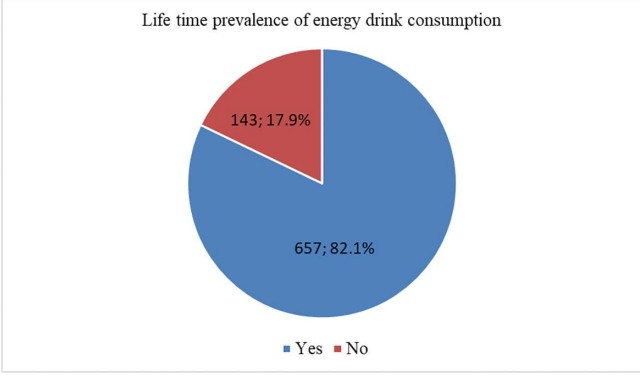

**Fig 1.  Lifetime prevalence of energy drink consumption among high school students in Erbil, Iraq.**

## Prevalence of energy drink consumption

Fig 2 displays the previous month's prevalence of energy drink consumption among high school students in Erbil, Iraq. More than half of the participants,57.6% (n=461), reported consuming energy drinks in the previous month, while 42.4% (n=339) had not consumed any during the same period. This finding indicates a very high current consumption rate, suggesting that energy drink consumption is common among high school students.

## Association of sociodemographic characteristics and prevalence of energy drink consumption

Table 2 shows the relationship between the previous month's energy drink consumption and the research sample's socio-demographic characteristics. The analysis included 800 participants; among them, 461 (57.6%) reported having consumed energy drinks in the month prior to data collection, whereas 339 did not.

The analysis found that there are significant associations between several socio-demographic factors. Notably, gender was significantly associated with energy drink consumption (p<0.001), with a higher proportion of male students (50.8%) reporting consumption compared to female students (49.2%). However, this difference is considered slight. Similarly, a statistically significant association (p=0.006) with socio-economic status was observed in the analysis, with higher energy drink consumption reported among participants from a medium socio-economic status background compared to both high (43.1%) and low (9.7%). Additionally, the education level of the head of the family demonstrated a statistically significant association with energy drink consumption (p<0.001). The head of the families with a primary education level who can read and write reported a higher consumption rate of 50.8% than those with other education levels. Moreover, living arrangement was significantly associated with energy drink consumption (p=0.023). Participants who live with single parent are more likely to consume energy drinks than those living with both parents, or in other living arrangements.

In contrast, no significant statistical associations were found between energy drinks and some variables, such as participant education level (p=0.270), Religion (p=0.929), Ethnicity (p=0.119), Family income (p=0.057), and occupation of the head of the family (p=0.353).

Prevalence of energy drink consumption

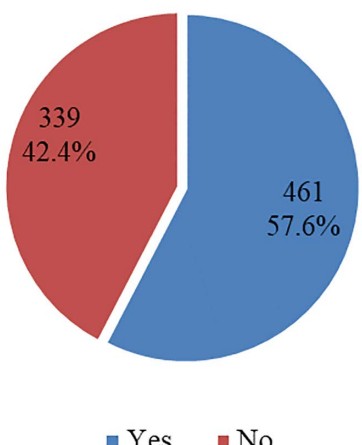

**Fig 2. Previous month prevalence of energy drink consumption among high school students in Erbil, Iraq.**

**Table 2. The association between the prevalence of energy drink consumption and socio-demographic variable among high school students, with chi square test result (p < 0.05).**

| | | Have you consumed any energy drinks in the previous month? | | | | P-value |
|---|---|---|---|---|---|---|
| | | Yes | | No | | |
| | | No. | % | No. | % | |
| **Academic level** | Grad 10 | 220 | 47.7 | 159 | 46.9 | 0.270 |
| | Grad 11 | 146 | 31.7 | 95 | 28.0 | |
| | Grad 12 | 95 | 20.6 | 85 | 25.1 | |
| **Sex** | Male | 234 | 50.8 | 108 | 31.9 | **<0.001** |
| | Female | 227 | 49.2 | 231 | 68.1 | |
| **Religion** | Muslim | 458 | 99.3 | 337 | 99.4 | 0.929 |
| | Christian | 2 | .4 | 1 | .3 | |
| | Kakayi | 1 | .2 | 1 | .3 | |
| **Living Arrangement** | With both parents | 433 | 93.9 | 327 | 96.5 | **0.023** |
| | with a single parent | 24 | 5.2 | 6 | 1.8 | |
| | Others | 4 | .9 | 6 | 1.8 | |
| **Ethnicity** | Kurdish | 439 | 95.2 | 316 | 93.2 | 0.119 |
| | Arabic | 14 | 3.0 | 9 | 2.7 | |
| | Turkish | 8 | 1.7 | 14 | 4.1 | |
| **Education level of the head of the family** | Illiterate | 28 | 6.1 | 16 | 4.7 | **<0.001** |
| | Can read and write, Primary | 234 | 50.8 | 119 | 35.1 | |
| | Intermediate school | 47 | 10.2 | 49 | 14.5 | |
| | High school or vocational | 31 | 6.7 | 47 | 13.9 | |
| | Bachelor's degree (college) and above | 121 | 26.2 | 108 | 31.9 | |
| **Family income** | Not sufficient | 75 | 16.3 | 65 | 19.2 | 0.057 |
| | Sufficient for daily needs | 265 | 57.5 | 166 | 49.0 | |
| | Exceeds for daily needs | 121 | 26.2 | 108 | 31.9 | |
| **Occupation of head of family** | Unskilled workers | 60 | 13.0 | 41 | 12.1 | 0.353 |
| | Lower professional | 304 | 65.9 | 212 | 62.5 | |
| | High professional | 97 | 21.0 | 86 | 25.4 | |
| **Socio-economic status** | Low | 47 | 10.2 | 33 | 9.7 | **0.006** |
| | Medium | 273 | 59.2 | 166 | 49.0 | |
| | High | 141 | 30.6 | 140 | 41.3 | |

P-values were calculated using the Chi-square test. Statistically significant association are indicated in bold (p < 0.05). Socio-economic status was categorized based the socioeconomic scoring system described in the method section.

These findings indicate that specific socio-demographic characteristics, particularly gender, the education level of the head of the family, living arrangement, and socio-economic status may be associated with the likelihood of energy drink consumption among high school students in Erbil city of the Kurdistan region of Iraq.

## Multivariate logistic regression analysis of socio-demographic characteristics associated with energy drink consumption

The result of the Chi-square test was used to determine the independent socio-demographic predictors of energy drink consumption within the previous month, and a multivariate logistic regression model was applied for this purpose. Variables included in the analysis model were gender, the education level of the head of the family, the living arrangement

of the participants, and socio-demographic status. This depends on the significance in the bivariate analysis (Chi-square test). The results of this analysis are summarized in Table 3.

The analysis revealed that the students' genders was a statistically significant predictor of energy drink intake. Male respondents were significantly more likely to report energy drink consumption compared to females (reference group), with an adjusted odds ratio (AOR) of 2.156 (95% CI: 1.595–2.916, p<0.001). This result suggested that male students had more than double the odds of consuming energy drinks compared to female students after controlling for other variables.

In addition to gender, living arrangement also demonstrated a statistically significant association with energy drink consumption. Participants who live with a single parent had significantly greater odds of consuming energy drinks compared to those living with both parents (reference group), with an AOR of 2.724 (95% CI: 1.075–6.900, p=0.035). Respondents who live with individuals other than their parents had lower odds of energy drink consumption (AOR=0.526), even though this association did not reach statistical significance (95% CI: 0.140–1.977, p=0.342).

In terms of the education level of the head of the family of the study sample, the result indicated that none of the categories showed a statistically significant association with energy drinks consumption when compared to the reference group (Bachelor's degree and above). Although students whose parents were illiterate had greater odds of consuming energy drinks (AOR=2.463, 95% CI: 0.610–9.942, p=0.205), this relationship was not statistically significant. Similarly, socio-economic status did not emerge as a significant predictor in the adjusted model. However, respondents belonging to the low SES category had lower odds of energy drinks consumption compared to those from a high SES background (reference group); the difference was not statistically significant (AOR=0.611, 95% CI: 0.181–2.066, p=0.428).

Overall, the multivariate model identified male sex and living with a single parent as the only statistically significant independent predictors of energy drink consumption in the study sample.

**Table 3. Multivariate logistic regression analysis of socio-demographic characteristics associated with energy drink consumption a among high school students in Erbil, Iraq.**

| Multivariate Logistic Regression | | B | P- value | AOR | 95% C.I. for OR | |
|---|---|---|---|---|---|---|
| | | | | | Lower | Upper |
| **Sex** | Male | 0.768 | **<0.001** | 2.156 | 1.595 | 2.916 |
| | Female | R | | | | |
| **Living Arrangement** | With both parents | R | | | | |
| | With a single parent | 1.002 | **0.035** | 2.724 | 1.075 | 6.900 |
| | Others | −0.642 | 0.342 | 0.526 | 0.140 | 1.977 |
| **Education level of the head of the family** | Illiterate | 0.902 | 0.205 | 2.463 | 0.610 | 9.942 |
| | Can read and write, Primary | 0.508 | 0.343 | 1.662 | 0.582 | 4.751 |
| | Intermediate school | −0.030 | 0.957 | 0.970 | 0.321 | 2.936 |
| | High school or vocational | −0.567 | 0.077 | 0.567 | 0.302 | 1.064 |
| | Bachelor's degree (college) and above | R | | | | |
| **Socio-economic status** | Low | −0.493 | 0.428 | 0.611 | 0.181 | 2.066 |
| | Medium | 0.056 | 0.912 | 1.058 | 0.393 | 2.845 |
| | High | R | | | | |
| | Constant | −0.220 | 0.143 | 0.803 | | |

Multivariate logistic regression analysis. B=regression coefficient; AOR=adjusted odds ratio; CI=confidence interval; R=reference category. Statistically significant results (p<0.05) are showed in bold. Statistical significance was set at p<0.05.

## Discussion

In this study, the participants' socio-demographic characteristics provided an important context for interpreting the outcomes of energy drink consumption among high school students in Erbil, Iraq. Females comprised a higher percentage of the study sample in comparison to males. This figure is consistent with a previous study, which argued that gender variation could be found in energy drink consumption [22]. However, these minor differences may be due to the higher number of female schools in this study's target population and to the probability-proportional-to-size sampling method. The socio-demographic characteristics of the study sample shows a relatively homogenous sample in terms of ethnicity, religion, and family income stability, with a slight variation in socio-economic status. These features of the sample are important for the generalizability of the research's findings. In addition, it may influence the effectiveness of intervention strategies aimed at reducing energy drink consumption among adolescents. Recent studies indicate that features of religiosity can be associated with eating attitudes and related behaviors in adolescent. At the same time, living arrangements and family structures have been associated with differences in soft drink consumption and dietary patterns among adolescents [23,24].

The findings of this study underscore a high lifetime prevalence of energy drink consumption among adolescents in Erbil, Iraq. This lifetime prevalence is higher than the estimated worldwide prevalence of 54.7% [25]. Additionally, it is higher than a similar study conducted in Germany, which stated a prevalence rate of 61.7% [26]. Given the high lifetime prevalence of energy drink, the recommendation for an educational program aimed at reducing energy drink consumption has strong potential to yield measurable public health benefits.

The significant findings of this study have estimated the prevalence of energy drink consumption, which was 57.6%, indicating that a considerable proportion of the participants had consumed energy drinks in the previous month. It also represents a widespread and frequent energy drink consumption among adolescents in Erbil, Iraq. This current prevalence is significantly higher than the worldwide prevalence over the past 30 days, at 21.6% [25]. Additionally, it is also greater than findings from a study conducted in Saudi Arabia among university students, which reported that 29.3% of the university students consumed energy drinks [27].

Regarding the relationship between the sample's socio-demographic variables and the previous month's prevalence of energy drink consumption. The association between gender and energy drink consumption, observed in chi-square analysis, persisted in the multivariate logistics regression, strengthens the robustness of the finding and suggesting that gender was an independent predictor of energy drink consumption. Male respondents reported greater rates of energy drink intake than females. This finding is consistent with prior research indicating gender differences in energy drink consumption, with males often consuming more. It may be associated with cultural, social, and behavioral factors, for example, cultural norms that discourage such behaviors for females, like females who drink in public being judged more negatively than men, being perceived as less respectable [22,28,29]. Moreover, the observed higher prevalence of energy drink consumption among males in this study is consistent with recent international evidence. For instance, a study indicates that across years of national surveillance in Spain, prevalence is consistently higher among male adolescents [30]. Similarly, a recent study by [31]reported a higher previous month prevalence of energy drink consumption among male adolescents in an extensive national survey in Chile. This pattern may be partially reflecting behavioral and psychosocial mechanisms, as recent research suggests that personality traits associated with stimulation-seeking and extraversion, which are closely related to the sensation-seeking and risk-taking behaviors, are associated with higher energy drink consumption. However, this association should be interpreted cautiously as a gender difference, as it is not consistently observed across various populations and time periods.

Additionally, the academic level of the head of the family of participants was associated with energy drink consumption in a bivariate level, and this association was statistically significant. After applying multivariate regression analysis to adjust for relevant confounders, the result did not support this association. However, this pattern is similar to a study in the United Arab Emirates, which reported that adolescents from parents with a higher educational background are generally associated with greater awareness of the health risks associated with energy drink consumption. [32].

Moreover, a statistically significant association was found between participants' living arrangements and energy drink consumption in both bivariate and multivariate regression analyses. It indicates that the living arrangement factor is independently associated with energy drink consumption behaviors, even after controlling for potential confounders. However, the wide range of confidence intervals suggests variability in the estimate, and further research with larger sample size are recommended to confirm this association. Participants living with a single parent reported higher energy drink consumption compared to those living with both parents. This result is aligning with studies' findings in the USA, Europe, and the UK, where students living with their parents reported consuming fewer energy drinks and nutrient-poor foods. In contrast this pattern differs from the study's findings, which indicated that students living with their parents reported higher consumption of energy drinks and nutrient-poor foods, including snacks, fast food, and sugary drinks [33]. Furthermore, students' socio-economic status was found to be statistically significantly associated with the prevalence of energy drink consumption. Although this association did not persist in the multivariate regression model, suggesting the absence of an independent effect. The use of the multivariate logistic regression in this study provided a more robust framework for interpreting the associations.

## Conclusion

The study provides considerable evidence on the consumption patterns of energy drinks among youth and adolescents in Erbil, Iraq. It also assesses the influence of sociodemographic factors on energy drink consumption. More than half of the participants reported having consumed energy drinks in the previous 30 days. Additionally, this study found that the prior-month prevalence of energy drink consumption was statistically significantly associated with males. Furthermore, it highlights the need to introduce appropriate and stricter regulations on the marketing and sale of energy drinks, particularly to adolescents and young adults to curb their widespread availability. The high prevalence of energy drinks among high school students underscores the crucial need for environmental control, policy measures, and educational intervention to reduce energy drink consumption among adolescents in Iraq.

## Limitations

The study provides valuable insights into the frequency and patterns of energy drink intake among Iraqi adolescents. The study's reliance on self-reported data may introduce recall and social desirability biases.

## Supporting information

**S1 File. Minimal data set (SPSS).** SPSS data file containing the data set used for statistical analysis.
(SAV)

**S2 File. Socio-economic scores.** Document describing the scoring method used to calculate socio-economic status of the participants.
(DOCX)

## Acknowledgments

I would like to acknowledge the academic and administrative staff of the community health department of the College of Medicine in Hawler Medical University and my colleagues for their support, and appreciate the staff of the Department of Statistics of the Directorate of Education of Erbil for their outstanding cooperation.

## Author contributions

**Conceptualization:** Samir Mahmood Othman.

**Data curation:** Soran Abdullah Al-Keji.

**Formal analysis:** Soran Abdullah Al-Keji.

**Funding acquisition:** Soran Abdullah Al-Keji.

**Investigation:** Soran Abdullah Al-Keji.

**Methodology:** Soran Abdullah Al-Keji, Samir Mahmood Othman.

**Project administration:** Soran Abdullah Al-Keji, Samir Mahmood Othman.

**Resources:** Soran Abdullah Al-Keji.

**Software:** Soran Abdullah Al-Keji.

**Supervision:** Samir Mahmood Othman.

**Validation:** Soran Abdullah Al-Keji.

**Visualization:** Soran Abdullah Al-Keji.

**Writing – original draft:** Soran Abdullah Al-Keji.

**Writing – review & editing:** Soran Abdullah Al-Keji, Samir Mahmood Othman.

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
