## [Decision Letter · Decision Letter 0]

26 Jan 2026

Dear Dr. Al-Keji,

Thank you for submitting your manuscript to PLOS ONE. After careful consideration, we feel that it has merit but does not fully meet PLOS ONE’s publication criteria as it currently stands. Therefore, we invite you to submit a revised version of the manuscript that addresses the points raised during the review process.

Please submit your revised manuscript by Mar 12 2026 11:59PM If you will need more time than this to complete your revisions, please reply to this message or contact the journal office at plosone@plos.org . A letter that responds to each point raised by the academic editor and reviewer(s). You should upload this letter as a separate file labeled 'Response to Reviewers'.A marked-up copy of your manuscript that highlights changes made to the original version. You should upload this as a separate file labeled 'Revised Manuscript with Track Changes'.An unmarked version of your revised paper without tracked changes. You should upload this as a separate file labeled 'Manuscript'.

We look forward to receiving your revised manuscript.

Kind regards,

Marwan Salih Al-Nimer, MD, PhD

Academic Editor

PLOS One

**Journal Requirements:**

2. In the ethics statement in the Methods, you have specified that verbal consent was obtained. Please provide additional details regarding how this consent was documented and witnessed, and state whether this was approved by the IRB.

3. We note that your Data Availability Statement is currently as follows:

“All relevant data are within the manuscript and its Supporting Information files.”

If your submission does not contain these data, please either upload them as Supporting Information files or deposit them to a stable, public repository and provide us with the relevant URLs, DOIs, or accession numbers. For a list of recommended repositories, please see https://journals.plos.org/plosone/s/recommended-repositories

4. Please upload a new copy of Figures 1 and 2 as the detail is not clear. Please follow the link for more information:  https://journals.plos.org/plosone/s/figures

Reviewers' comments:

Reviewer's Responses to Questions

**Comments to the Author**

1. Is the manuscript technically sound, and do the data support the conclusions?

Reviewer #1: Yes

Reviewer #2: Yes

Reviewer #3: Yes

2. Has the statistical analysis been performed appropriately and rigorously?

Reviewer #1: Yes

Reviewer #2: No

Reviewer #3: No

3. Have the authors made all data underlying the findings in their manuscript fully available?

Reviewer #1: Yes

Reviewer #2: No

Reviewer #3: No

4. Is the manuscript presented in an intelligible fashion and written in standard English?

Reviewer #1: Yes

Reviewer #2: Yes

Reviewer #3: Yes

Reviewer #1: Generally, authors need to emphasize on standardization of terms/words.

Avoid using different words/terms but referring to the same thing.

Captions for table and figure must be specific.

Please improve the tables and figures as commented in the PDF.

Overall, the writing is fine, simple but able to highlight the findings.

Reviewer #2: The authors have made an effort to address a research gap in this manuscript. However, several points should be addressed.

1. The authors mention that the questionnaire used in this study was designed for this manuscript, which implies that it was developed by the authors. However, in another section, it is stated that the questionnaire was modified from previous studies. This point needs clarification. It is suggested that the authors report how many items were included for each topic and describe what the questions were. If available, information on the validity and reliability of the questionnaire should be provided.

Furthermore, it is not clear how the information on religion and living arrangements relates to the study objectives. The cutoff points used to determine socioeconomic status and family income should also be clarified, particularly as the questionnaire was completed by students without permission or confirmation from parents.

2. It is not clear how Figures 1 and 2 were developed, as no information related to this is provided in the statistical methods section.

3. In the Discussion section, gender variation could be explained in more detail. If gender was considered in the analysis, further analysis using adjusted associations is suggested, rather than relying only on chi-square tests.

Reviewer #3: This manuscript speaks to an important and timely public health issue, which is energy drink consumption among adolescents in an area where data are scarce. The study thoughtfully uses a suitable cross-sectional design with a sizable sample and employs standard statistical methods to explore how sociodemographic factors relate to energy drink use. This is a very relevant topic, and the findings provide some helpful initial insights for the Kurdistan Region of Iraq. Just a few areas related to methodology, analysis, ethics, and presentation could be improved to make sure the manuscript is perfectly polished and ready for publication.

Strengths

This study sheds light on important aspects of energy drink use among high school students in Erbil City. The sample size has been carefully chosen using established epidemiological methods, and the use of multistage cluster sampling helps ensure the data truly reflects the student population. The manuscript provides a comprehensive look at both lifetime and recent (30-day) consumption rates, giving a clear picture of usage trends. Also, ethical approval was obtained from an institutional ethics committee, ensuring the study was conducted responsibly.

Major Comments

1. Study design and interpretation

• The cross-sectional design restricts the ability to draw causal conclusions. However, the manuscript at times suggests causation, such as when sociodemographic factors are described as 'influencing' consumption. All interpretations should be clearly stated as associations rather than cause–and–effect links. Additionally, the Discussion section would benefit from clearly distinguishing findings supported directly by data from those that are more speculative, based on cultural or social considerations.

2. Statistical analysis

• The statistical analysis relies solely on bivariate chi-square tests. Although it is appropriate for preliminary evaluation, this approach ignores potential confounding variables. Additionally, the absence of effect sizes like odds ratios or risk ratios limits the ability to gauge the practical significance of the statistically significant results.

3. Ethical considerations

• Not obtaining written parental consent for minors requires ethical justification. Since participants are under 18, the manuscript should clarify if this approach aligns with local and international research ethics standards and whether assent procedures were employed.

• The claim that students completed the questionnaire “consciously” is unclear and needs further clarification.

4. Measurement and questionnaire

• The manuscript offers limited details on the validity and reliability of the revised questionnaire. It remains unclear if the instrument was pretested or piloted within the target population. Essential variables like “socioeconomic status” need clear definitions, including the methods used for measurement or classification.

Minor Comments

1. Language and clarity

• The manuscript features many grammatical errors, awkward phrasing, and typos that hinder readability. Repetition of words is frequent, especially in the Introduction and Discussion sections. Additionally, several sentences are overly lengthy and need to be simplified to improve clarity.

2. Formatting and presentation

• The manuscript contains formatting inconsistencies such as misplaced line numbers, irregular spacing, and table alignment problems. Additionally, table titles and footnotes need revision to enhance clarity and consistency. Figures should also be clearly labeled and consistently referenced in the text.

3. References

• Some references are outdated, and including more recent literature (from the last 5–7 years) could enhance the background and discussion.

**Do you want your identity to be public for this peer review?** For information about this choice, including consent withdrawal, please see our Privacy Policy

Reviewer #1: No

Reviewer #2: No

Reviewer #3: No

---

## [Author Response · Author response to Decision Letter 1]

19 Feb 2026

Dear Editor and Reviewers,

We sincerely thank the editor and reviewers for their careful evaluation of our manuscript and for their constructive comments. All suggestions were carefully considered and addressed in a point-by-point manner. The manuscript has been revised accordingly to improve its clarity, methodological rigor, and overall quality. Reviewer comments are reproduced below, followed by our responses, and corresponding changes have been incorporated into the revised manuscript.

General Requirements

Comment 1:

Response:

All article style requirements, including the specified file naming conventions, have been carefully followed.

Comment 2:

In the ethics statement in the Methods, you have specified that verbal consent was obtained. Please provide additional details regarding how this consent was documented and witnessed, and state whether this was approved by the IRB.

Response:

Ethical approval was obtained from the Institutional Ethics Committee of the College of Medicine Ethics Committee at Hawler Medical University, which serves as the official body responsible for reviewing research involving human participants. The committee approved the use of verbal informed consent due to the minimal-risk nature of the study. The consent process was explained verbally, witnessed by a designated school official, and documented in a consent log sheet. These details have been clarified in the revised Methods section.

Comment 3:

We note that your Data Availability Statement is currently as follows:

“All relevant data are within the manuscript and its Supporting Information files.”

Response:

3- All raw data necessary to replicate the study results have now been uploaded.

Comment 4:

Please upload a new copy of Figures 1 and 2 as the detail is not clear.

Response:

4- Revised versions of Figures 1 and 2 with improved clarity have now been uploaded.

Review Comments to the Author

Reviewer #1:

Comment 1:

Generally, authors need to emphasize on standardization of terms/words. Avoid using different words/terms but referring to the same thing.

Response:

The manuscript has been revised to ensure consistent use of terminology throughout.

Comment 2:

Captions for table and figure must be specific. Please improve the tables and figures as commented in the PDF.

Response:

Table and figure captions have been made more specific, and the tables and figures have been improved in accordance with the reviewer’s comments provided in the PDF.

Reviewer #2:

Comment 1:

The authors mention that the questionnaire used in this study was designed for this manuscript, which implies that it was developed by the authors. However, in another section, it is stated that the questionnaire was modified from previous studies. This point needs clarification. It is suggested that the authors report how many items were included for each topic and describe what the questions were. If available, information on the validity and reliability of the questionnaire should be provided.

Response:

We would like to clarify that the questionnaire was developed for a broader research project; however, this manuscript reports only the subset of items relevant to its objectives. The questionnaire was developed by the authors based on items adapted and modified from previous studies, rather than being entirely newly designed. In addition, the number of items included under each topic and a brief description of the questionnaire content have now been provided in the Methods section. Information on questionnaire validity and reliability has also been added, including expert review for content validity, pilot testing.

Comment 2:

It is not clear how the information on religion and living arrangements relates to the study objectives. The cutoff points used to determine socioeconomic status and family income should also be clarified, particularly as the questionnaire was completed by students without permission or confirmation from parents.

Response:

The manuscript has been revised to clarify the relevance of religion and living arrangements to the study objectives.

Comment 3:

It is not clear how Figures 1 and 2 were developed, as no information related to this is provided in the statistical methods section.

Response:

3- The manuscript has been revised to explain the development of Figures 1 and 2 in the Statistical Analysis section. These figures were generated from the uploaded raw dataset using the analytical procedures described.

Comment 4:

In the Discussion section, gender variation could be explained in more detail. If gender was considered in the analysis

Response:

Gender was included in the analysis, and gender-related variation was considered when interpreting the findings. The Discussion addresses possible explanations for observed differences, including behavioral and sociocultural factors, supported by relevant literature.

Comment 5:

further analysis using adjusted associations is suggested, rather than relying only on chi-square tests.

Response:

A multivariate logistic regression analysis was conducted to examine adjusted associations between socio-demographic factors and energy drink consumption while controlling for potential confounders. Adjusted results have been added to the Results section (Table 3) and interpreted in the Results and Discussion sections.

Reviewer #3:

Comment 1:

The cross-sectional design restricts the ability to draw causal conclusions. However, the manuscript at times suggests causation, such as when sociodemographic factors are described as 'influencing' consumption. All interpretations should be clearly stated as associations rather than cause–and–effect links.

Response:

1- We fully agree that the cross-sectional design of the study does not allow causal inferences. The manuscript has been revised to ensure that interpretations are framed strictly in terms of associations rather than cause–and–effect relationships. Specifically, causal language such as “influencing,” “leading to,” or “determinants” has been replaced with non-causal terms including “associated with.

Comment 2:

The discussion section would benefit from clearly distinguishing findings supported directly by data from those that are more speculative, based on cultural or social considerations.

Response:

The Discussion section has been revised to more clearly distinguish findings that are directly supported by the study data from interpretations that are speculative and based on cultural or social considerations.

Comment 3:

The statistical analysis relies solely on bivariate chi-square tests. Although it is appropriate for preliminary evaluation, this approach ignores potential confounding variables. Additionally, the absence of effect sizes like odds ratios or risk ratios limits the ability to gauge the practical significance of the statistically significant results.

Response:

We performed a multivariate logistic regression analysis to assess adjusted associations while controlling for potential confounding variables. Effect size measures, including adjusted odds ratios (AORs) with 95% confidence intervals, have now been reported to enhance the interpretation of the findings. The multivariate logistic regression results have been added to the Results section (Table 3), and the interpretation has been incorporated into both the Results and Discussion sections.

Comment 4:

Not obtaining written parental consent for minors requires ethical justification. Since participants are under 18, the manuscript should clarify if this approach aligns with local and international research ethics standards and whether assent procedures were employed.

Response:

The manuscript has been revised to clarify the rationale for not obtaining written parental consent and to explicitly describe the assent procedures used for minor participants. As this was a minimal-risk, school-based study, written parental consent was not required under local ethical guidelines. In addition, verbal assent was obtained from all students prior to data collection after providing clear, age-appropriate information about the study. These procedures were reviewed and approved by the College of Medicine Ethics Committee at Hawler Medical University and are consistent with relevant international research ethics standards. The Ethics section has been updated accordingly.

Comment 5:

The claim that students completed the questionnaire “consciously” is unclear and needs further clarification.

Response:

The term “consciously” has been removed and replaced with a clearer description of the questionnaire administration process, including the instructions provided to students and the voluntary nature of participation.

Comment 6:

The manuscript offers limited details on the validity and reliability of the revised questionnaire. It remains unclear if the instrument was pretested or piloted within the target population.

Response:

Additional details on the validity and reliability of the revised questionnaire have now been included in the Methods section. The instrument was pretested through a pilot study conducted within the target population, and the results were used to refine the questionnaire prior to data collection.

Comment 7:

Essential variables like “socioeconomic status” need clear definitions, including the methods used for measurement or classification.

Response:

Socioeconomic status has now been clearly defined in the revised manuscript's Methods section.

Comment 8:

The manuscript features many grammatical errors, awkward phrasing, and typos that hinder readability. Repetition of words is frequent, especially in the Introduction and Discussion sections. Additionally, several sentences are overly lengthy and need to be simplified to improve clarity.

Response:

The manuscript has been carefully revised and thoroughly proofread multiple times to improve grammatical accuracy, eliminate awkward phrasing and repetition, and enhance overall clarity. Several lengthy sentences, particularly in the Introduction and Discussion sections, have been simplified to improve readability.

Comment 9:

The manuscript contains formatting inconsistencies such as misplaced line numbers, irregular spacing, and table alignment problems. Additionally, table titles and footnotes need revision to enhance clarity and consistency. Figures should also be clearly labeled and consistently referenced in the text.

Response:

All formatting inconsistencies have been carefully corrected in the revised manuscript, including line numbering, spacing, and table alignment. Table titles and footnotes have been revised to ensure clarity and consistency, and all figures have been clearly labeled and consistently referenced in the text.

Comment 10:

Some references are outdated, and including more recent literature (from the last 5–7 years) could enhance the background and discussion.

Response:

The reference list has been carefully reviewed, and most applicable references have been updated to include more recent literature from the last 5–7 years. Relevant contemporary studies have been incorporated into both the Introduction and Discussion sections to strengthen the scientific background and contextualization of the findings.

---

## [Editor Report · Decision Letter 1]

24 Feb 2026

Prevalence and consumption patterns of energy drinks among Iraqi adolescents: A Cross-sectional study

PONE-D-26-00648R1

Dear Dr. ,SORAN ABDULLAH Al-Keji

We’re pleased to inform you that your manuscript has been judged scientifically suitable for publication and will be formally accepted for publication once it meets all outstanding technical requirements.

Kind regards,

Marwan Salih Al-Nimer, MD, PhD

Academic Editor

PLOS One
---

## [Editor Report · Acceptance letter]

PONE-D-26-00648R1

PLOS One

Dear Dr. Al-Keji,

I'm pleased to inform you that your manuscript has been deemed suitable for publication in PLOS One. Congratulations! Your manuscript is now being handed over to our production team.

Kind regards,

on behalf of

Professor Marwan Salih Al-Nimer

Academic Editor

PLOS One